# Are There Any Differences in the Healing Capacity between the Medial Collateral Ligament’s (MCL) Proximal and Distal Parts in the Human Knee? Quantitative and Immunohistochemical Analysis of CD34, α-Smooth Muscle Actin (α-SMA), and Vascular Endothelial Growth Factor (VEGF) Expression Regarding the Epiligament (EL) Theory

**DOI:** 10.3390/biomedicines12030659

**Published:** 2024-03-15

**Authors:** Georgi P. Georgiev, Yordan Yordanov, Lyubomir Gaydarski, Richard Shane Tubbs, Łukasz Olewnik, Nicol Zielinska, Maria Piagkou, Julian Ananiev, Iva N. Dimitrova, Svetoslav A. Slavchev, Ivan Terziev, Athikhun Suwannakhan, Boycho Landzhov

**Affiliations:** 1Department of Orthopedics and Traumatology, University Hospital Queen Giovanna-ISUL, Medical University of Sofia, 1527 Sofia, Bulgaria; 2Department of Pharmacology, Pharmacotherapy and Toxicology, Faculty of Pharmacy, Medical University of Sofia, 1000 Sofia, Bulgaria; yyordanov@pharmfac.mu-sofia.bg; 3Department of Anatomy, Histology and Embryology, Medical University of Sofia, 1431 Sofia, Bulgaria; lgaidarsky@gmail.com (L.G.); landzhov_medac@abv.bg (B.L.); 4Department of Anatomical Sciences, St. George’s University, St. George 1473, Grenada; shane.tubbs@icloud.com; 5Department of Neurosurgery, Tulane University School of Medicine, New Orleans, LA 70112, USA; 6Department of Neurology, Tulane University School of Medicine, New Orleans, LA 70112, USA; 7Department of Structural and Cellular Biology, Tulane University School of Medicine, New Orleans, LA 70112, USA; 8Department of Surgery, Tulane University School of Medicine, New Orleans, LA 70112, USA; 9Department of Anatomical Dissection and Donation, Medical University of Lodz, 90-419 Lodz, Poland; lukasz.olewnik@umed.lodz.pl (Ł.O.); nicol.zielinska@stud.umed.lodz.pl (N.Z.); 10Department of Anatomy, School of Medicine, National and Kapodistrian University of Athens, 11527 Athens, Greece; piagkoumara@gmail.com; 11Department of General and Clinical Pathology, Faculty of Medicine, Trakia University, 6000 Stara Zagora, Bulgaria; operation@abv.bg; 12Department of Cardiology, University Hospital “St. Ekaterina”, Medical University of Sofia, 1431 Sofia, Bulgaria; dimytrova@yahoo.com; 13University Hospital of Orthopedics “Prof. B. Boychev”, Medical University of Sofia, 1614 Sofia, Bulgaria; s.slavchev@medfac.mu-sofia.bg; 14Department of Pathology, University Hospital Queen Giovanna-ISUL, 1527 Sofia, Bulgaria; titia@abv.bg; 15Department of Anatomy, Faculty of Science, Mahidol University, Bangkok 10400, Thailand; athikhun.suw@mahidol.edu; 16In Silico and Clinical Anatomy Research Group (iSCAN), Bangkok 10400, Thailand

**Keywords:** medial collateral ligament, epiligament, theory, knee joint

## Abstract

The human knee is a complex joint that comprises several ligaments, including the medial collateral ligament (MCL). The MCL provides stability to the knee and helps prevent its excessive inward movement. The MCL also has a thin layer of connective tissue known as the epiligament (EL), which adheres to the ligament. This unique feature has drawn attention in the field of ligament healing research, as it may have implications for the recovery process of MCL injuries. According to the EL theory, ligament regeneration relies heavily on the provision of cells, blood vessels, and molecules. The present study sought to compare the expression of vascular endothelial growth factor (VEGF), CD34, and α-smooth muscle actin (α-SMA) in healthy knees’ proximal and distal MCL segments to better understand how these proteins affect ligament healing. By improving the EL theory, the current results could lead to more effective treatments for ligament injury. To conduct the present analysis, monoclonal antibodies were used against CD34, α-SMA, and VEGF to examine samples from 12 fresh knee joints’ midsubstance MCLs. We identified a higher cell density in the EL than in the ligament connective tissue, with higher cell counts in the distal than in the proximal EL part. CD34 immunostaining was weak or absent in blood vessels and the EL, while α-SMA immunostaining was strongest in smooth muscle cells and the EL superficial layer. VEGF expression was mainly in the blood vessels’ tunica media. The distal part showed more SMA-positive microscopy fields and higher cell density than the proximal part (4735 vs. 2680 cells/mm^2^). Our study identified CD34, α-SMA, and VEGF expression in the MCL EL, highlighting their critical role in ligament healing. Differences in α-SMA expression and cell numbers between the ligament’s proximal and distal parts may explain different healing capacities, supporting the validity of the EL theory in ligament recovery.

## 1. Introduction

The medial collateral ligament (MCL) plays a crucial role in stabilizing the knee joint. It runs along the inner side of the knee, connecting the femur to the tibia. MCL injuries are prevalent, accounting for approximately 90% of knee injuries. Along with the anterior cruciate ligament (ACL), the MCL is one of the most frequently injured ligaments in the knee joint [1,2]. In the last decade, the number of MCL sports injuries has risen [3,4,5]. Although the MCL shows excellent healing capacities, a healed MCL has structural and mechanical properties inferior to those of the uninjured ligament [6]. The Epiligament (EL) theory provides a fascinating explanation for the process of ligament regeneration. The theory suggests that the EL, which is a tissue layer surrounding the ligament, plays a crucial role in this process. The EL is rich in fibroblasts, progenitor cells, blood vessels, and connective tissue cells, which are essential components for the regeneration of ligaments. When a ligament is injured, these cells migrate from the EL toward the injured area, targeting the regeneration of the ligament. This unique mechanism highlights the importance of EL in the process of ligament repair and regeneration [7,8,9,10,11,12,13].

Fibroblasts are specialized cells that play a vital role in the healing process of ligaments. They produce a variety of proteins that contribute significantly to this process, including collagen matrix metalloproteinases, fibromodulin, decorin, and fibronectin. These proteins are responsible for breaking down and regenerating the injured ligament, making them crucial for successful healing [7,8,9,10,11,12,13,14,15]. By gaining a deeper understanding of the EL molecular properties, we can enrich our knowledge of the complex mechanisms involved in ligament healing and potentially develop more effective treatments for this type of injury.

CD34 is a cell surface marker protein primarily linked with the endothelial progenitor cells and is involved in repairing injured tissues [16,17]. Blood vessel walls are rich reservoirs of CD34-expressing stem cells [16,18]. The contractile α-smooth muscle actin (α-SMA) protein is commonly used as a marker for smooth muscle cells [19] and myofibroblasts [20]. When exposed to inflammatory mediators post-injury, fibroblasts can undergo a transition to proto-myofibroblasts and subsequently transform into typical myofibroblasts characterized by α-SMA de novo expression [21]. This expression significantly enhances contractile myofibroblast activity [19]. Menetrey et al. [22] identified α-SMA-positive cells in the MCL, migrating towards the lesion’s center three days post-injury. Vascular endothelial growth factor (VEGF) is essential for angiogenesis [23] and ligament healing [24].

The current study aims to enhance the understanding of the EL theory by examining the expression of crucial molecules (VEGF, CD34, and α-SMA) in ligament healing, following an immunohistochemical (IHC) analysis of the MCL EL. This research seeks to establish whether there are any discrepancies in cell count and molecule expression between the ligament’s proximal and distal parts, and how these alterations could impact the ligament’s ability to heal. Ultimately, this study will provide valuable insights into the role the EL plays in the healing process.

## 2. Materials and Methods

### 2.1. Tissue Preparation

This study involved histological and IHC assessments of samples from the proximal and distal parts of the MCL in 12 fresh European cadavers. These cadavers were chosen based on their age, with a mean age of 55 years (range 49–62), and gender, including 7 females and 5 males. None of the cadavers had any clinical data indicating knee osteoarthritis or any scars from previous knee surgery. A skin incision was made to collect the samples, and the underlying subcutaneous tissue was carefully dissected to expose the MCL and the EL outer surface. This ensured that the samples were taken from the intended area with precision and accuracy. The collected samples were then subjected to further analysis. Samples were taken from the ligament’s proximal and distal thirds and underwent a standard fixation procedure [25].

### 2.2. Light Microscopy

The specimens were meticulously prepared for light microscopy using a Leica microtome (Wetzlar, Germany). By cutting 5 μm thick sections, the samples were mounted on microscope slides and stained with hematoxylin and eosin, following established methods. These steps were taken to ensure the highest level of accuracy and precision in the resulting images, allowing for a thorough analysis of the specimens under examination [11,12].

### 2.3. Immunohistochemistry (IHC)

Several specimens were 10% formalin-fixed, embedded in paraffin, and then sliced to 4 μm thickness. This study employed a set of antibodies from DAKO Cytomation, Agilent (Glostrup, Hovedstaden, Denmark) which included monoclonal mouse anti-human α-SMA (M0851), monoclonal mouse anti-human VEGF antibody (M7273), and monoclonal mouse anti-human CD34 (M7165). These antibodies were diluted to a ratio of 1:100 for better specificity and sensitivity. For the detection of these antibodies, the EnVision™ FLEX+, Mouse, and High pH (Link) detection system (K8002) (DAKO Agilent) was utilized. The experiment involved the implementation of controls in eighteen (18) sections. To capture representative IHC staining fields, an Olympus CX21 microscope was used, which was fitted with an Olympus C5050Z digital camera manufactured by Olympus Optical Co., Ltd., Tokyo, Japan. The combination of these two devices was instrumental in ensuring accurate and reliable results.

### 2.4. Semiquantitative Analysis

The expressions of CD34, VEGF, and α-SMA were semiquantitatively analyzed using ImageJ 1.53f51 software, which is a widely used open-source image processing program that can be downloaded for free from the official website (http://imagej.nih.gov/ij/) [13]. This software allows for the measurement of pixel intensities, which can be used to quantify the expression levels of different proteins in IHC-stained samples. To assess the staining intensity, we utilized the IHC Profiler plugin, which is a free download available from the official ImageJ website (https://sourceforge.net/projects/ihcprofiler/) [14]. This plugin allows for the automatic and objective scoring of IHC-stained samples based on a four-tier system, which includes high positive (3+), positive (2+), low positive (1+), and negative (0) classifications. The scoring is based on the intensity and proportion of stained cells in each visual field. To ensure accuracy, our study involved analyzing at least ten random visual fields on each of the five slides. The final score was determined by calculating the average score of all visual fields. This approach allowed us to obtain a more comprehensive and representative assessment of the staining intensity and expression levels of CD34, VEGF, and α-SMA in our samples.

### 2.5. Cell Numbers

We determined the density of cells using a sophisticated method based on supervised machine learning in Ilastik software (stable release 1.3.3/2019) [26]. A pixel classifier was trained to accurately identify the tissue area, background, and nuclei in the image. We then processed the segmentation masks of the nuclei to identify particles, filtered out any noise and debris, and normalized the cell counts to the tissue area masks of the same image. The resulting cell counts were stated as the number of cells present in each square millimeter (mm^2^) of the tissue. This technique allowed us to obtain precise and reliable information on cell density, which is crucial for understanding cellular- and tissue-level processes.

### 2.6. Statistical Analysis

To analyze the image count data in a detailed manner, the statistical programming language R v4.2.2 [26] was utilized along with the integrated development environment R Studio v2023.03.0 + 386 [27]. For comparing groups, a *t*-test was employed, assuming similar dispersions. To ensure the reliability and accuracy of the findings, statistical significance was considered when *p* < 0.05. To make the data more visually understandable, the ggplot2 v3.4.0 package [28] was used to generate graphic representations that effectively convey the analysis results.

## 3. Results

### 3.1. Light Microscopic Observations

The EL was identified as similar proximally (Figure 1a,b) and distally (Figure 2a,b). It consisted of active and non-active fibroblasts, adipose cells, and extracellular collagen fibers, either alone or in groups. Most of the neurovascular bundles of the EL–ligament complex were located in the EL. The EL had more fibroblasts at both ends than in the middle. The morphology of the EL was remarkably similar to that of the synovium.

### 3.2. Expression of CD34, α-SMA, and VEGF in the MCL EL

Immunostaining for CD34 in the endothelial layers of blood vessels was weak or absent in the MCL proximal and distal parts. This was observed in the EL, in contrast to the midsubstance (Figure 3a,b and Figure 4a,b). Immunostaining for α-SMA was strongest in the smooth muscle cells of the blood vessels tunica media, and in the EL superficial layer (Figure 3c,d and Figure 4c,d), as compared to the midsubstance. Positive VEGF IHC staining was localized mostly in the blood vessels’ tunica media (Figure 3e,f and Figure 4e,f), in contrast to the midsubstance, where a positive reaction was detected only in the blood vessels endothelial layer.

To gain insight into the intensities of CD34, α-SMA, and VEGF IHC reactions in the proximal and distal sections of the MCL, Table 1 presents an overview. Our analysis relied on two key parameters: the proportion of images with a specific overall score (as visualized in Figure 5) and the average percentage of image areas with corresponding scores across all experimental group images.

Upon examination of the images, it was observed that the distribution of CD34 staining with DAB showed similar patterns in both the proximal and distal sections of the MCL EL. The majority of fields, approximately 95%, were negative. Conversely, VEGF DAB staining showed no significant differences between the proximal and distal regions. Although approximately 85% of the microscopy fields were negative, the percentages of low positive and positive fields were several-fold higher than those of the images stained with plate number 1. The overall score of all images stained with VEGF DAB was negative. In contrast, α-SMA DAB staining revealed the highest percentage of positive microscopy fields overall, with 39.2% for the distal and 26.4% for the proximal parts (excluding 0), compared to VEGF (less than 20%) and CD34 (less than 10%) DAB staining. Additionally, α-SMA DAB staining showed the most pronounced differences between the proximal and distal parts. The number of positive microscopy fields overall was greater in the distal part, where the number of negative fields was lowest (about 60%). This was confirmed by the image distribution with overall IHC scores, which were negative in the proximal part and all low positive in the distal part (Figure 5). The mean cell counts in the EL MCL distal part were significantly higher than those in the proximal part (4735 vs. 2680 cells/mm^2^, p 1.3 × 10^−6^) (Figure 6). In conclusion, based on the analysis of the images, the distribution of CD34, VEGF, and α-SMA staining varied in different regions of the MCL EL. While CD34 exhibited similar distributions between the proximal and distal parts, VEGF showed no marked differences, and α-SMA showed the highest percentage of positive microscopy fields overall, with the most pronounced differences between the proximal and distal parts.

## 4. Discussion

The current study seeks to examine the expressions of CD34, α-SMA, and VEGF in the MCL concerning EL theory and to explore their implications for ligament healing. Given that the ligament is an avascular structure, it is believed to have limited to no potential for healing. In contrast, the EL is vascularized and contains a diverse range of cells, including fibroblasts, fibrocytes, adipocytes, and mast cells. These cells produce vital compounds, such as collagens, matrix metalloproteinases, and fibronectin, which are essential for maintaining ligament homeostasis. The present findings are relevant as they provide valuable insights into the mechanisms underlying ligament healing. By elucidating the role of specific cells and compounds in this process, this study contributes to the development of more effective treatment strategies for ligament injuries. Furthermore, it highlights the importance of the EL in the context of ligament healing, suggesting that this structure may hold the key to unlocking the full healing potential of the ligament [8,9,10,11,12,13,15].

The enlisted cells are involved in the complex mechanisms of degradation, remodeling, and proliferation that occur in an injured ligament. Fibroblasts are implicated in phagocytosis, differentiation, and collagen synthesis [8,9,10,13]. Countless collagen fibers, dispersed in all directions, were also observed in the EL. One hypothesis suggests that single or grouped collagen fibers respond to ligament tension [13]. The adipocytes located within the extracellular matrix (ECM) function as exceptional packing material [13]. Notably, the principal neurovascular bundles are mainly situated in the EL of the MCL [8,9,10,11,12,13,15]. The EL is paramount in maintaining normal ligamental growth and homeostasis.

Georgiev and colleagues [10] conducted a rodent study on MCL injury healing, revealing that EL cells migrate into the gap caused by torn ligaments, connecting the injured ends and promoting the healing process. The authors additionally proved that the essential collagens needed for ligament repair primarily reside within the MCL EL [11,15].

CD34 expression within the EL suggests the presence of stem/progenitor cells that are essential for the regenerative processes involved in ligament recovery and repair [16,17]. Mifune et al. [29] demonstrated in a rat model that cells positive for CD34 lead to enhanced collagen II production and increased angiogenesis, thus improving healing. The blood vessel walls provide a substantial source of stem/progenitor cells expressing CD34 surface markers [16,18]. CD34 cells circulating in the bloodstream exhibit robust vasculogenic activity in the MCL injury, significantly contributing to the ligament’s healing. Preclinical findings illustrate morphological changes during healing accompanied by concurrent vasculogenesis. Tei and colleagues [30] revealed that circulating CD34+ cells have shown potential for future clinical applications in the healing and remodeling of ligaments. This finding sheds light on the possibility of using these cells to optimize the recovery process and improve the outcomes of ligament-related injuries in patients. Our results revealed that in the EL of the MCL, CD34 was expressed very weakly or not at all in the blood vessels’ endothelial layers and the EL. The analyzed images showed similar distributions of microscopy fields overall, with most of them (about 95%) being negative in both proximal and distal parts. The EL in the MCL midpart is considered to be the primary donor of CD34+ cells, with a low positive expression of CD34. The EL in the MCL midpart is involved in fostering enhanced healing potential. The MCL proximal and distal parts have worse healing capacity regarding CD34 expression [25].

Myofibroblasts have diverse origins, yet their development follows a consistent sequence [20]. Following injury, they are exposed to inflammatory stimuli, and mechanical microenvironment forces are reported to adopt the ‘proto myofibroblast’ phenotype, subsequently transforming into typical myofibroblasts, which are characterized by α-SMA de novo expression [21]. Increased α-SMA expression facilitates a substantial enhancement of myofibroblast contractile activity [20]. Menetrey et al. [22] have reported that α-SMA-positive cells emerged in the MCL, within three days post-injury, and subsequently migrated towards the lesion’s center. The restoration of the MCL’s original length and in situ strain is accomplished by myofibroblasts [31]. Our research has shown that α-SMA-positive reactions are mainly present in the smooth muscle cells of blood vessels in the tunica media and the superficial layer of the MCL EL.

Expression of α-SMA-positive cells, including myofibroblasts within the MCL, highlights the crucial significance of the EL in ligament healing. SMA DAB staining showed the most pronounced differences between the proximal and distal parts of the MCL EL. There were more positive microscopy fields in the distal part, where the number of negative fields was lowest. The α-SMA expression in the distal part of the MCL EL and the higher number of cells in this part suggest that, according to the EL theory, the distal part has a greater healing capacity.

EGF plays a critical role in the activation, migration, and proliferation of endothelial cells in various pathologies [32]. It is especially important during the early phases of proliferation and remodeling, where it acts as a powerful angiogenic stimulator [33]. VEGF receptors are located exclusively on endothelial cells and are expressed in developing blood vessels [34]. VEGF is recognized as the most significant promoter of vascular growth, and it directly regulates endothelial cell behaviors like migration, proliferation, and differentiation [35]. By stimulating angiogenesis, VEGF helps to facilitate access to the healing site, although the benefits of increased neovascularization for clinical outcomes remain unclear [36]. Acting as an endogenous stimulator, VEGF contributes to both angiogenesis and heightened vascular permeability [37]. Various studies have shown that VEGF production peaks after the inflammatory phase during the natural healing of injured ligaments and tendons [38,39,40]. Reports indicate that VEGF and blood vessel formation reach their peak between five- and nine days post-injury [14,33]. Elevated levels of angiogenic growth factors, including VEGF, at the injury site, are associated with a well-defined pattern of vascular ingrowth from both the epi- and intra-tendinous blood supplies to the repair site. This neovascularization progresses along the epitenon’s surface through a typically avascular region, supplying extrinsic cells, nutrients, and growth factors to the injured area [33]. As an endothelial mitogen, VEGF promotes angiogenesis, enhances capillary permeability, and contributes to fibrous integration between the tendon and bone during the early postoperative stage [41].

Wei et al. proposed that VEGF plays a crucial role in ligament healing by promoting angiogenesis and accelerating remodeling [24]. In our study, we found that VEGF was mostly expressed in the tunica media of the blood vessels. Therefore, we confirm the validity of the EL theory and reiterate that EL is the primary supplier of blood vessels while VEGF contributes to ligament healing. Our analysis of VEGF DAB staining showed no significant differences between the proximal and distal parts of the MCL EL.

Following injury, ligaments heal through the formation of scar tissue rather than regeneration [42,43,44]. Several studies indicated that the EL serves as a primary source of connective tissue cells that contribute to scar tissue formation during ligament healing [13,42,45,46,47,48]. It is established that fibroblasts, which are critical in scar formation, are mobile cells capable of migrating from the EL to the injured ligament [13,45,47,49,50]. Ligament injuries prompt the release of various cell types from the EL, including neutrophils and mitotic cells, up to the fifth day after injury [14]. This suggests a bilateral interaction between the EL and ligament, indicating that they collaborate in facilitating effective healing. The MCL proximal injury tends to heal with less remaining laxity and a higher likelihood of stiffness than injury at the distal end [51,52].

***Study Strengths:*** The findings of the current study demonstrate slightly higher expression of CD34 and VEGF in the MCL EL distal part and a higher α-SMA expression in the proximal part. Such discrepancies could explain the disparity in healing between the MCL EL proximal and distal parts.

The present study has several limitations.

The cadavers’ age could potentially introduce bias due to age-related changes [53]. In response, recently deceased cadavers with a mean age of 55 years, free from osteoarthritis or trauma, were used.The subjective nature of visual quantification of IHC images exposes them to considerable inter- and intra-observer variability. The IHC Profiler plugin integrated with ImageJ software was implemented to mitigate this problem.Only healthy MCL EL was investigated.

## 5. Conclusions

The present study delves into the expression of CD34, α-SMA, and VEGF in the human MCL EL and its critical role in ligament healing. The EL serves as a crucial source of blood vessels to the MCL–EL complex, which is vital for ligament nutrition and recovery. The distal part of the ligament shows a higher number of cells and α-SMA expression, which could explain its superior healing potential. CD34 expression reveals that the MCL’s proximal and distal parts have a lower capacity for stem cell migration into the ruptured site, according to the EL theory. VEGF expression in both the distal and proximal parts confirms that the EL is the primary blood vessel donor and is involved in ligament healing. These findings extend the existing EL theory and provide insights into the intricate healing processes of the MCL. Further research is needed to explore the EL’s significant role in MCL healing.

## Figures and Tables

**Figure 1 biomedicines-12-00659-f001:**
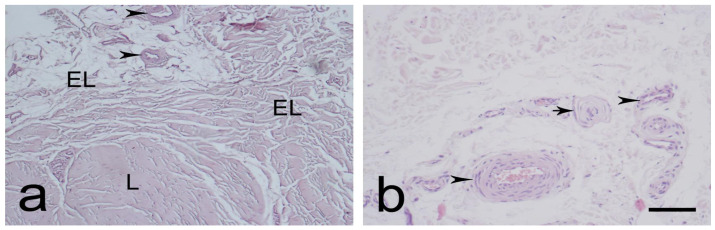
Typical morphology of the proximal part of the medial collateral ligament (MCL) epiligament (EL), (L)-ligament (**a**,**b**). Black arrowheads—EL blood vessels; black arrows—sensory nerve endings. Hematoxylin and eosin stain. Scale bar 100 µm; (**b**) Scale bar 50 µm.

**Figure 2 biomedicines-12-00659-f002:**
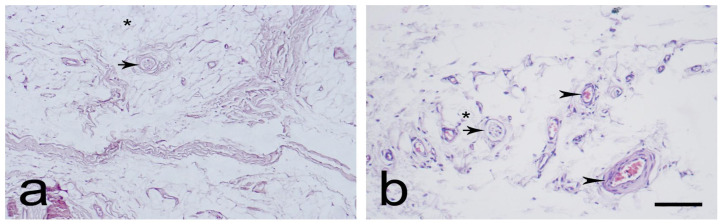
Typical morphology of the distal part of the medial collateral ligament (MCL) epiligament (EL) (**a**,**b**). Black arrowheads—EL blood vessels; black asterisks—adipocytes; and black arrows—sensory nerve endings. Hematoxylin and eosin stain. Scale bar 100 µm.

**Figure 3 biomedicines-12-00659-f003:**
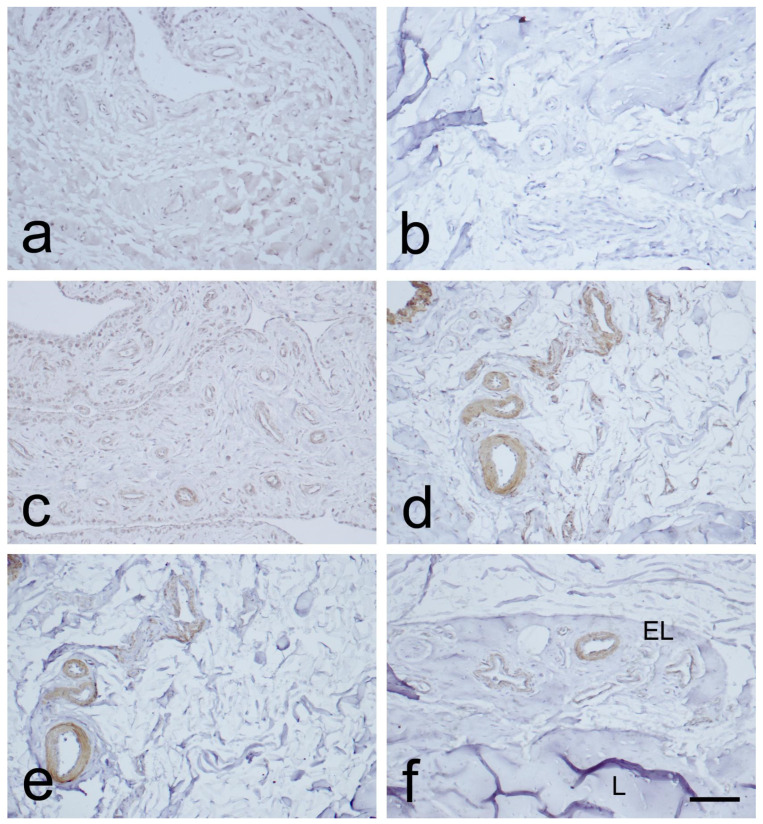
Immunohistochemical (IHC) staining for CD34, α-SMA, and VEGF in the epiligament (EL) of the medial collateral ligament (MCL) proximal part. (**a**,**b**): CD34 IHC staining. (**c**,**d**): α-SMA IHC staining. (**e**,**f**): VEGF IHC staining. Scale bar 100 µm.

**Figure 4 biomedicines-12-00659-f004:**
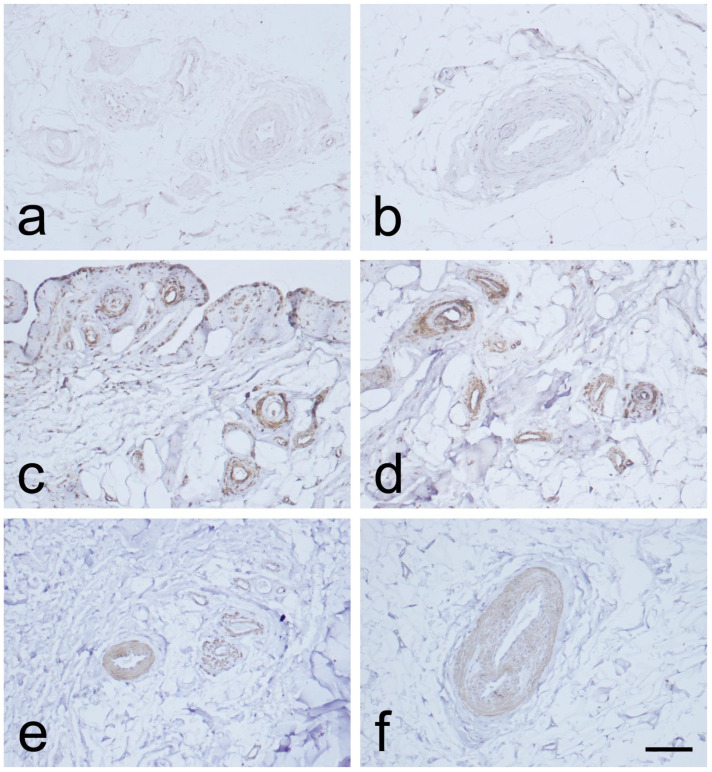
Immunohistochemical (IHC) staining for CD34, α-SMA, and VEGF in the epiligament (EL) of the distal part of the medial collateral ligament (MCL) of the knee. (**a**,**b**): CD34 IHC staining. (**c**,**d**): α-SMA IHC staining. (**e**,**f**): VEGF IHC staining. Scale bar 100 µm.

**Figure 5 biomedicines-12-00659-f005:**
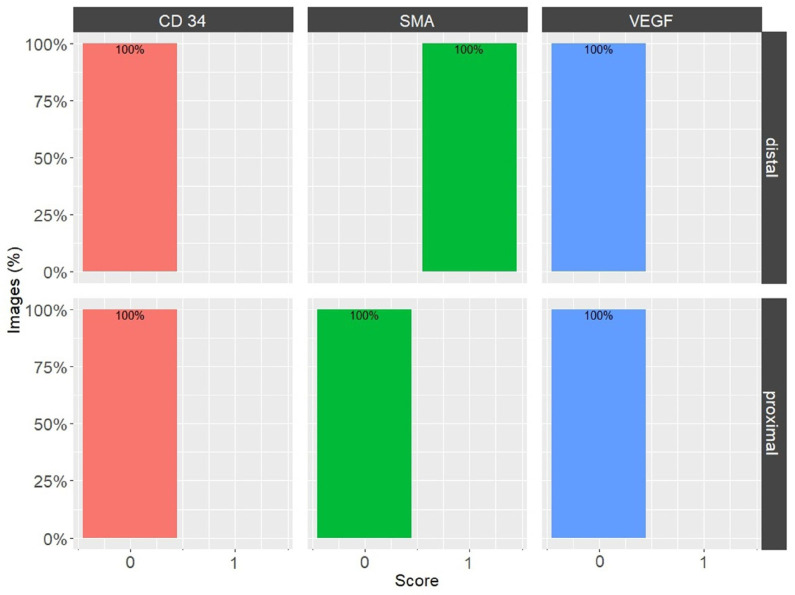
Percentages of overall image grades, for CD34, α-SMA, and VEGF in the EL of the medial collateral ligament (MCL) proximal and distal parts. Bars are color-coded: CD34—orange, α-SMA—green, and VEGF—blue.

**Figure 6 biomedicines-12-00659-f006:**
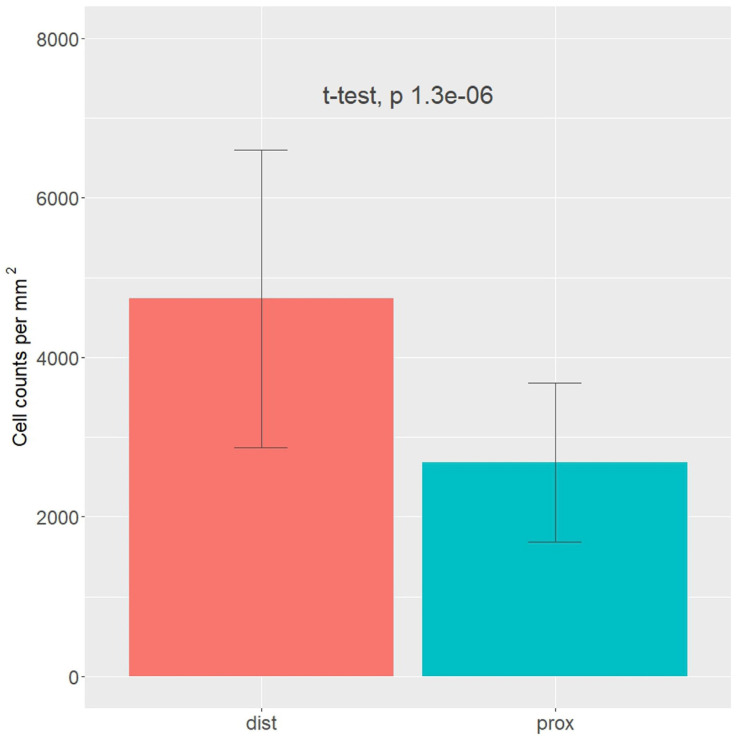
Cell density in the epiligament (EL) of the medial collateral ligament (MCL) proximal (blue color) and distal (red color) parts based on nuclei counts and normalized by the total tissue area fraction in the image. *t*-test with *p*-value presented above the bars.

**Table 1 biomedicines-12-00659-t001:** Semiquantitative analysis of the immunohistochemical (IHC) expression of VEGF, CD34, and α-SMA in the proximal and the distal parts of the epiligament (EL) of the medial collateral ligament (MCL). The percentage for each score represents the percentage of visual fields that the IHC Profiler assigned this score to.

IHC Marker	Distal Part of the MCL EL		Proximal Part of the MCL EL	
	Score	%	Score	%
**VEGF**	High Positive (3+)	0.1	High Positive (3+)	0.1
	Positive (2+)	2.3	Positive (2+)	2.1
	Low Positive (1+)	14.5	Low Positive (1+)	14.6
	Negative (0)	83.1	Negative (0)	83.2
**CD34**	High Positive (3+) (0.0%)	0.0	High Positive (3+)	0.1
	Positive (2+)	0.4	Positive (2+)	0.7
	Low Positive (1+)	3.8	Low Positive (1+)	6.1
	Negative (0)	95.9	Negative (0)	93.0
**α-SMA**	High Positive (3+)	2.5	High Positive (3+)	0.2
	Positive (2+)	12.8	Positive (2+)	5.1
	Low Positive (1+)	23.0	Low Positive (1+)	21.0
	Negative (0)	61.8	Negative (0)	73.6

## Data Availability

The raw data supporting the conclusions of this article will be made available by the authors upon request.

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
