# Peer review of "Are There Any Differences in the Healing Capacity between the Medial Collateral Ligament’s (MCL) Proximal and Distal Parts in the Human Knee? Quantitative and Immunohistochemical Analysis of CD34, α-Smooth Muscle Actin (α-SMA), and Vascular Endothelial Growth Factor (VEGF) Expression Regarding the Epiligament (EL) Theory"

_biomedicines, 2024, doi:10.3390/biomedicines12030659_

Round 1
Reviewer 1 Report
Comments and Suggestions for Authors
The authors describe some features of the importance of the epiligament of the medial collateral ligament in healing capacity. They carried out immunohistochemical analyses in order to quantify some molecules involved in this process. The work scientifically sounds, appears well written and the results are convincing. Probably the figures could be improved because the immunohistochemical stain seems pale and it could be more evident. The major criticism in this work is the quantification of the staining. The authors made a subjective evaluation "by eye". They could use a common digital software applied on figures to exactly quantify the color intensity.
Another little thing. In the lines 114-115 there is a Bulgarian phrase, probably written as a note by an author, which he forgot to delete but which perhaps indicates somethings disturbing. Was the antibody not present in the market perhaps used? The authors should explain this note that they forgot to delete. Among 13 authors, there was not one who noticed the presence of a phrase to delete? Has the work been checked by everyone?
Author Response
Response to Reviewers: We appreciate the constructive feedback provided by the reviewers and have addressed each point in detail:
Response to Reviewer 1:
- The immunohistochemical analyses utilized objective software evaluation methods. Each visual field underwent automatic scoring based on a four-tier system using the validated algorithm by Varghese et al. (2014) through the IHC Profiler plugin for ImageJ. Additionally, cell densities were assessed through an algorithm involving cell nuclei segmentation using Ilastik software, followed by cell counting and density evaluation via ImageJ. Standardized algorithms were applied to all sample images to ensure an objective evaluation approach.
- The oversight regarding the Bulgarian phrase in lines 114-115 has been rectified in the revised manuscript. Apologies for the inconvenience.
In conclusion, we believe these revisions significantly improve the quality and readability of the manuscript while maintaining its scientific rigor and integrity. We appreciate the thorough review process and the opportunity to address the reviewers' comments. We are confident that the revised manuscript meets the standards of Biomedicines.
Sincerely,
Georgi P. Georgiev

Reviewer 2 Report
Comments and Suggestions for Authors
biomedicines-2882000
The article is a very interesting analysis of the immunohistochemical properties of the MCL epiligament.
I have though several comments to make.
The title is somehow misleading because the authors did not study the healing capacity of the MCL but only the histochemical properties of the epiligament. The title should be amended to better represent the topic evaluated.
Abstract
Line 33, no comma after knee
Line 37, the MCL proximal and distal segments. Epiligament not MCL.
line 38. The healing of the MCL was not evaluated in this study.
line 39. Samples from the mid-substance. Describe the exact location of sample harvesting and correct throughout the paper.
Line 42 The cell density. Please provide numbers when you refer to measurements.
Introduction
Line 56. knee joint is. Are
Line 61, according to it. Please delete.
Line 63, targeting its regeneration. Please improve wording.
Lines 72-77. Not necessary in the introduction.
Line 81. The. Not necessary
Line 86-87. and whether these differences affect the healing capacities of the ligament. This was not studied.
Line 92. ‘fresh European cadavers’. Please provide race and details regarding the cause of death, the timing of tissue harvesting after death and the preservation of the cadaver in the meantime. Delayed tissue harvesting may have significantly affected the histochemical properties of the tissue. Please provide information and comment.
Line 98. the proximal and distal thirds of the ligament. Describe the harvesting location in more detail and keep throughout the paper.
Line 114-115. _(_само _антитялото _за _V_E_G_F_ _го _няма,_ _но _добавих _това:_ _от). I guess this sentence is not essential. Otherwise translate it.
Line 137. Statistical analysis report should be improved. “using a t-test, assuming similar dispersions”. There are valid tests to evaluate normal distribution of data.
The complete lack of inter- or intraobserver variability is the main fault of this paper.
3. Results
3.1. Light microscopic observations
If you are using quantitative terms such as most or more, you should provide numbers.
Line 150-151. The EL morphology was strikingly similar to that of the syno vium. This should be the opening sentence
Line 214. ‘p 1.3e-06’. Please rewrite and correct in Figure 6 as well.
Discussion
You should start with the main findings of the study.
Please stick to the analysis of your data in the discussion.
Line 224. and mast cells. Please provide reference.
Line 228. ligament. Please provide reference.
Line 234.’ The facts presented here demonstrate that EL is paramount in maintaining normal 234 ligamental growth and homeostasis.’ This statement is not supported by the results of your study.
Line 244. ‘Mifune et al.’ Please provide reference.
Line 300. to Wei et al. Please provide reference.
Lines 306-316. This paragraph could be placed in the begging of the discussion.
Conclusions
Present only the findings of your study without assumptions and provide guidelines about future research directions.
Comments on the Quality of English Languageminor editing is needed
Author Response
Response to Reviewers: We appreciate the constructive feedback provided by the reviewers and have addressed each point in detail:
Response to Reviewer 2:
- We have addressed and incorporated all provided suggestions regarding the better readability of our manuscript.
- Abstract: We have provided cell density values where indicated.
- Methods: We have opted not to provide additional detail on the cadavers and tissue preparation methods in the current study, as these aspects have been extensively described in our previous works, which we have cited accordingly. This decision was made to minimize duplication with our prior publications and to maintain focus on the novel aspects of the current study.
- Methods and results: About the statistical suggestions: “Statistical analysis report should be improved. “using a t-test, assuming similar dispersions”. There are valid tests to evaluate normal distribution of data. The complete lack of inter- or intraobserver variability is the main fault of this paper.” - We did not test this assumption because we used Welch's variation of the t-test, which is robust in handling differences in variance and deviations from normality. Additionally, the use of automated software algorithms to obtain IHC scores and cell density values removes the need for testing inter- or intraobserver variability since the results are not direct observations. Therefore, we do not see this suggestion as appropriate.
- Results: “If you are using quantitative terms such as most or more, you should provide numbers.” - We acknowledge that including the values in brackets would enhance clarity, and thus, we have added the numeric values in brackets for clarification when discussing the distribution of IHC scores among fields.
- Discussion: we rewrote the discussion for clarity in accordance with the provided suggestions. Moreover, the requested references were provided.
In conclusion, we believe these revisions significantly improve the quality and readability of the manuscript while maintaining its scientific rigor and integrity. We appreciate the thorough review process and the opportunity to address the reviewers' comments. We are confident that the revised manuscript meets the standards of Biomedicines.
Sincerely,
Georgi P. Georgiev

Reviewer 3 Report
Comments and Suggestions for Authors
The Authors aimed to assess and compare the VEGF, CD34, and α-SMA expression in the MCL proximal and distal segments in healthy human knees to elucidate the proteins’ effect on ligament healing outcomes.
The topic is interesting and the study well designed. The paper is well prepared.
I have a few minor concerns.
Please define proximal and distal portion of MCL. Also, the preparation of fresh cadavers must be detailed.
Was previous knee trauma an exclusion criteria for cadavers selection?
Semiquntitative evaluation should be further described.
Figure 5 legend not clear.
Table 1 disorganized. Also, a statistical analysis should be added.
Author Response
Response to Reviewers: We appreciate the constructive feedback provided by the reviewers and have addressed each point in detail:
Response to Reviewer 3:
- Definitions for the proximal and distal segments of the MCL have been provided. Detailed explanation of the tissue preparation process has been provided in our previous works, which we have cited. In order to minimize the duplication rate with our previous publications, we decided to not further expand on this matter.
- The results of the semiquantitative evaluation have been presented in a manner that emphasizes the outcomes relevant to the scientific question of the study. This semiquantitative evaluation method was developed by Varghese et al. (2014), who extensively validated it. Detailed procedures for this method are available as a reference in their publication.
- Clarifications have been made to the legend for Figure 5, and the organization of Table 1 has been improved. Statistical analysis has been included in Figure 6, where numeric data for cell counts is presented. However, in Figure 5, fractions of analyzed images with specific IHC scores are depicted. Since the analyzed slides exhibited homogeneous staining and consistent overall scores, there was no variability in this parameter. Therefore, we deemed it inappropriate to add statistical analyses in Figure 5 to evaluate whether differences are due to chance.
In conclusion, we believe these revisions significantly improve the quality and readability of the manuscript while maintaining its scientific rigor and integrity. We appreciate the thorough review process and the opportunity to address the reviewers' comments. We are confident that the revised manuscript meets the standards of Biomedicines.
Sincerely,
Georgi P. Georgiev

Reviewer 4 Report
Comments and Suggestions for Authors
The study proves intriguing, and I am inclined to support its publication. However, I have several suggestions for improvement:
Clarification of the study's hypothesis is essential. What underlies the authors' belief in the disparate healing abilities of various sessions of the medial collateral ligament?
Explicitly state that this is a cadaveric study and confirm that Institutional Review Board (IRB) approval is unnecessary.
Provide a detailed explanation of the cadaver preparation process. The authors should expound on the methods employed in preparing the cadavers for experimentation.
Address the absence of statistical analysis in the study. A robust statistical approach is crucial for ensuring the validity and reliability of the findings.
In Figure 6, elucidate the meaning of the bars. A clear explanation of what each bar represents will enhance the reader's understanding of the data presented.
Author Response
Response to Reviewers: We appreciate the constructive feedback provided by the reviewers and have addressed each point in detail:
Response to Reviewer 4:
- We have clarified the study's hypothesis and explicitly stated that this is a cadaveric study.
- A detailed explanation of the tissue preparation process has been described in our previous studies, which we have cited.
- Further elucidation of the meaning of the bars in Figure 6 has been provided for enhanced reader understanding.
- To ensure the validity and reliability of our findings, we employed an objective automated software approach for image analysis. Additionally, we utilized Welch’s t-test statistic for evaluating the numeric cell density values. These methods were chosen for their ability to provide robust and accurate results, enhancing the credibility of our study.
In conclusion, we believe these revisions significantly improve the quality and readability of the manuscript while maintaining its scientific rigor and integrity. We appreciate the thorough review process and the opportunity to address the reviewers' comments. We are confident that the revised manuscript meets the standards of Biomedicines.
Sincerely,
Georgi P. Georgiev
